# Self-Assembling Graph Perceptrons

**Jialong Chen**[1]✉**, Tong Wang**[1,2]✉**, Bowen Deng**[1]✉**, Luonan Chen**[3,4]✉**,**
**Zibin Zheng**[1]✉**, Chuan Chen**[1]*✉

[1]Sun Yat-sen University, [2]Texas A&M University,
[3]Shanghai Jiao Tong University, [4]University of Chinese Academy of Sciences

## Abstract

Inspired by the workings of biological brains, humans have designed artificial neural networks (ANNs), sparking profound advancements across various fields. However, the biological brain possesses high plasticity, enabling it to develop simple, efficient, and powerful structures to cope with complex external environments. In contrast, the superior performance of ANNs often relies on meticulously crafted architectures, which can make them vulnerable when handling complex inputs. Moreover, overparameterization often characterizes the most advanced ANNs. This paper explores the path toward building streamlined and plastic ANNs. Firstly, we introduce the Graph Perceptron (GP), which extends the most fundamental ANN, the Multi-Layer Perceptron (MLP). Subsequently, we incorporate a self-assembly mechanism on top of GP called Self-Assembling Graph Perceptron (SAGP). During training, SAGP can autonomously adjust the network's number of neurons and synapses and their connectivity. SAGP achieves comparable or even superior performance with only about 5% of the size of an MLP. We also demonstrate the SAGP's advantages in enhancing model interpretability and feature selection.

## 1 Introduction

With the exceptional intelligence of the brain, humanity has created a remarkable modern civilization. This intelligence stems from the brain's intricate structure, resulting from long-term environmental adaptation and natural selection. Research shows that the survival environment profoundly impacts shaping the brain, which in turn indirectly shapes human cognitive and emotional abilities [1].

On the other hand, since the 1940s, researchers have been exploring how to simulate the behavior of the biological brain using computers to build artificial intelligence agents. It was only in recent decades that a framework known as artificial neural networks (ANNs) has indeed demonstrated significant potential in this field. ANNs simulate neurons and their synaptic connections in the brain, determining each neuron's output by the strength of the input signals they receive.

Despite the powerful capabilities of ANNs, their design seems to deviate from the initial intent of mimicking the biological brain. In ANNs, the number of neurons and synaptic connection patterns are predefined, resulting in noticeable vulnerabilities when the network faces complex environments. [2] and [3] have demonstrated, from theoretical and experimental perspectives, respectively, the significant impact of the number of hidden layers and neurons on the ability of Multi-Layer Perceptrons (MLPs). Moreover, modern deep models are often over-parameterized, with the most advanced large language models reaching a parameter scale of trillions. In contrast, biological brains continually self-assemble throughout their lifecycle, developing remarkable capabilities. For instance, a nematode can manage all its behaviors with fewer than 500 neurons [4].

---

*Corresponding author.

39th Conference on Neural Information Processing Systems (NeurIPS 2025).

Recently, some studies have focused on creating ANNs that can assemble themselves based on input without relying on prior knowledge. [5] and [6] proposed learning genomes that regulate neuronal behavior, enabling self-adjustment of synaptic connection rules without altering the number of neurons. The neural development program (NDP, [7, 8]) first suggests regulating neuron growth through genomes, allowing the network to develop from a single neuron and incrementally add new neurons and synapses in response to inputs, eventually growing into a network of a predefined size. However, NDP does not fully realize biological self-assembly. It neglects the most important reason biological neural systems maintain their efficiency and compactness: neuronal apoptosis [9]. Furthermore, since NDP explicitly encodes a genome for generating the network structure and optimizes it by reinforcement learning, the high computational cost prevents it from running on even a typical-scale dataset.

In this paper, we first introduce a generalized version of the Multi-Layer Perceptron (MLP), namely the Graph Perceptron (GP). Building upon this, we present the **S**elf-**A**ssembling **G**raph **P**erceptrons (SAGP)—the first model with *full* self-assembly capability. SAGP begins from the simplest form and autonomously determines when to grow or undergo apoptosis during its developmental cycle. Synaptic connections between neurons are dynamically adjusted. Unlike previous works, SAGP does not explicitly model the genome that controls neuronal behavior and topology. Instead, it achieves self-assembly by establishing assembly rules and simulating the competitive pressures found in nature. This makes SAGP more aligned with the general paradigm of modern deep networks and results in a significant improvement in the assembly speed by more than 10,000 times. We demonstrate that SAGP can achieve a more streamlined perceptron topology than MLP while highlighting the potential of the self-assembly mechanism in enhancing model interpretability and feature selection.

## 2 Related work

**Neural network bionics**

Scientists have long sought inspiration from the biological world to develop algorithms for solving real-world problems. Early bio-inspired strategies, such as evolutionary and genetic algorithms [10, 11], solve complex optimization problems by simulating the process of biological evolution. Meanwhile, artificial neural networks, which simulate the process of neuronal interactions through synaptic connections [12], have

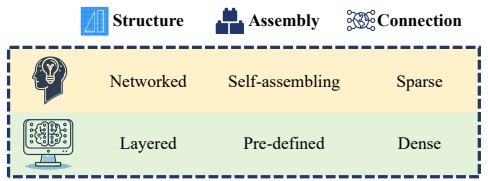

Figure 1: Biological NN v.s. Artificial NN.

achieved great success in numerous applications. However, ANNs, represented by MLP, are predefined as layered structures with fixed sizes and usually require a large number of neurons and dense synaptic connections for optimal performance, which is contrary to the properties of biological neural networks (Fig. 1). Consequently, a growing body of research is focused on designing networks with more bio-inspired features. Spiking Neural Networks (SNNs) mimic the mechanism of neurons transmitting information through action potentials, enabling asynchronous and low-power information processing [13, 14]; Liquid Neural Networks (LNNs) simulate the neural connection of the nematodes, using a small number of neurons to generate continuous-time outputs [15]. Plastic Neural Networks (PNNs) emulate the developmental processes of organisms, dynamically adjusting their structures based on environmental states [16].

**Synaptic-level plasticity** The characteristic of plastic ANNs lies in their ability to adjust their structure in response to environmental changes. Initially, this research area focused on synaptic-level plasticity; inspired by early neurobiological theories, researchers developed mechanisms such as the Hebbian rule [17] and Spike-Timing Dependent Plasticity (STDP, [18]) to enable ANNs to self-regulate synaptic strength. These classical methods are unsupervised, while more modern approaches employ backpropagation to learn synaptic rules [19, 20] or use meta-learning to obtain genomes that control synaptic behavior [5, 6].

**Neuron-level plasticity** Some methods allow neurons to adjust their state based on the environment, such as by modifying weights [21], changing activation functions [22], or learning rates [23]. Neuroevolution [24, 25] encodes the network topology as individuals in a population and finds the optimal ones by evolution. [26] considers parameter pruning based on plastic neurons. Only recently

has the concept of self-assembly — the dynamic increase of neurons within a single network (as opposed to a population) — been introduced by Neural Developmental Program (NDP, [7, 8]). We achieve a fully self-assembling model with neural competition and apoptosis mechanisms. Compared to NDP, our model improves the assembly speed by over $10^4$ times per epoch and shows the ability to integrate with modern deep models. See Appendix A for more related works.

# 3 Self-Assembling Graph Perceptrons

This section introduces our approach, the Self-Assembling Graph Perceptrons (SAGP), in detail. Without additional constraints, the free connections between neurons may exhibit a more general connection pattern than the layered connections, namely, a graph-structured connection. In Section 3.1, we explore how information is updated in a graph-structured perceptron model (i.e., GP), which forms the foundation for achieving the network's self-assembly capability. Section 3.2 introduces the approach by which GP achieves self-assembly through establishing assembly rules and simulating competitive pressures.

## 3.1 From MLPs to graph perceptrons

Since the inception of perceptron models, they have been conventionally regarded as layered structures [12, 27, 28, 29]. Although fully connected perceptron models like Hopfield Networks [30] and Boltzmann Machines [31] emerged in the past, they gradually became marginalized due to practical limitations. However, we re-examine this concept inspired by the message-passing mechanism [32].

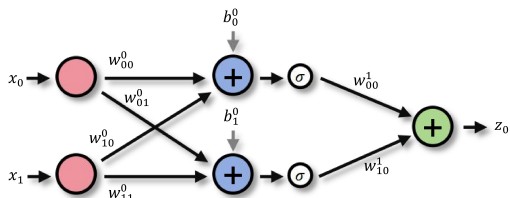

Figure 2: A simple example of a Multi-Layer Perceptrons, where $\sigma$ represents the activation function.

Take a simple MLP with 2 input neurons, 2 hidden neurons in 1 hidden layer, and 1 output neuron as an example (Fig. 2). Let $\boldsymbol{x} = \begin{bmatrix} x_0 \\ x_1 \end{bmatrix}$ and $\boldsymbol{z} = [z_0]$ represent the model's input and output, respectively, then we have:

$$\boldsymbol{z} = \boldsymbol{W}_1^T \cdot \sigma(\boldsymbol{W}_0^T \cdot \boldsymbol{x} + \boldsymbol{b}_0), \tag{1}$$

where $\boldsymbol{W}_0 = \begin{bmatrix} w_{00}^0 & w_{01}^0 \\ w_{10}^0 & w_{11}^0 \end{bmatrix}$, $\boldsymbol{b}_0 = \begin{bmatrix} b_0^0 \\ b_1^0 \end{bmatrix}$, and $\boldsymbol{W}_1 = \begin{bmatrix} w_{00}^1 \\ w_{10}^1 \end{bmatrix}$.

Now, we number all neurons sequentially in the order of input, hidden, and output layers. In this way, the topology of the MLP and the weights of its edges can be represented by a $5 \times 5$ adjacency matrix $\boldsymbol{A}$. If the $i$-th neuron has a synaptic connection to the $j$-th neuron, then $A_{ij}$ is the weight of that synapse; otherwise, $A_{ij} = 0$. Input neurons carry self-loops, meaning they continuously send information to the network. We also construct a vector $\boldsymbol{b}$ of length 5 to represent the biases of all the neurons, where the biases of input and output neurons are set to 0, as follows:

$$\boldsymbol{A} = \begin{bmatrix} \boldsymbol{I}_{2\times 2} & \boldsymbol{W}_0 & \boldsymbol{0}_{2\times 1} \\ \boldsymbol{0}_{2\times 2} & \boldsymbol{0}_{2\times 2} & \boldsymbol{W}_1 \\ \boldsymbol{0}_{1\times 2} & \boldsymbol{0}_{1\times 2} & \boldsymbol{0}_{1\times 1} \end{bmatrix}, \quad \boldsymbol{b} = \begin{bmatrix} \boldsymbol{0}_2 \\ \boldsymbol{b}_0 \\ \boldsymbol{0}_1 \end{bmatrix}. \tag{2}$$

Here, $\boldsymbol{0}$ represents a zero matrix or vector. If we input $\boldsymbol{x}$ at the input neurons and use zero inputs for other neurons, then, after one step of message-passing [32] on the graph described by $\boldsymbol{A}$, we obtain the same intermediate results at the hidden neurons as we would in MLPs:

$$\sigma \left( \boldsymbol{A}^T \cdot \begin{bmatrix} \boldsymbol{x} \\ \boldsymbol{0}_2 \\ \boldsymbol{0}_1 \end{bmatrix} + \boldsymbol{b} \right) = \begin{bmatrix} \sigma(\boldsymbol{x}) \\ \sigma(\boldsymbol{W}_0^T \boldsymbol{x} + \boldsymbol{b}_0) \\ \boldsymbol{0}_1 \end{bmatrix}. \tag{3}$$

By repeating this process once again, the output neuron produces the same result as the output of the MLPs:

$$\boldsymbol{A}^T \cdot \begin{bmatrix} \sigma(\boldsymbol{x}) \\ \sigma(\boldsymbol{W}_0^T \boldsymbol{x} + \boldsymbol{b}_0) \\ \boldsymbol{0}_1 \end{bmatrix} + \boldsymbol{b} = \begin{bmatrix} \sigma(\boldsymbol{x}) \\ \boldsymbol{W}_0^T \sigma(\boldsymbol{x}) + \boldsymbol{b}_0 \\ \boldsymbol{z} \end{bmatrix}. \tag{4}$$

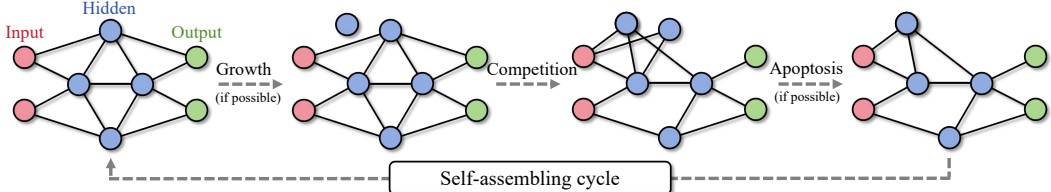

Figure 3: A schematic of our model (SAGP). It assembles itself by simulating neuron growth, competition, and apoptosis.

Thus, we obtain an alternative representation of $z$, where we use Python-style indexing $[\texttt{-2:}]$ to denote taking the last two elements of a vector to form a new one:

$$z = \left( A^T \cdot \sigma \left( A^T \cdot \begin{bmatrix} x \\ 0_3 \end{bmatrix} + b \right) + b \right)_{[-2:]}. \tag{5}$$

It is easy to show that similar conclusions hold for MLPs of any number of hidden layers and sizes (See appendix B).

Based on this observation, we now define a generalized perceptron model, referred to as the **G**raph **P**erceptrons (**GP**). Let the sets of input neurons, hidden neurons, and output neurons be denoted as $\mathcal{I}$, $\mathcal{H}$, and $\mathcal{O}$, respectively. GP has the following adjacency matrix and biases:

$$A = \begin{bmatrix} I_{|\mathcal{I}| \times |\mathcal{I}|} & A_{\mathcal{I} \to \mathcal{H}} & A_{\mathcal{I} \to \mathcal{O}} \\ 0_{|\mathcal{H}| \times |\mathcal{I}|} & A_{\mathcal{H} \to \mathcal{H}} & A_{\mathcal{H} \to \mathcal{O}} \\ 0_{|\mathcal{O}| \times |\mathcal{I}|} & 0_{|\mathcal{O}| \times |\mathcal{H}|} & 0_{|\mathcal{O}| \times |\mathcal{O}|} \end{bmatrix}, \; b = \begin{bmatrix} 0_{|\mathcal{I}|} \\ b_{\mathcal{H}} \\ 0_{|\mathcal{O}|} \end{bmatrix}. \tag{6}$$

GP allows direct connections from input neurons to output neurons and interconnections between any two hidden neurons. In GP, the feature update process is described as:

$$\forall l \in [0, L-1], \; h^{l+1} = A\widehat{x}^l + b, \; \widehat{x}^{l+1} = \sigma(h^{l+1}) \text{ where } \widehat{x}^0 = \begin{bmatrix} x \\ 0_{|\mathcal{H}| + |\mathcal{O}|} \end{bmatrix} \text{ and } z = h^L_{[-|\mathcal{O}|:]}. \tag{7}$$

When $A_{\mathcal{I} \to \mathcal{O}} = 0$, $A_{\mathcal{H} \to \mathcal{H}} = I$, and $L = 2$, the GP degenerate to MLPs with 1 hidden layer. $L$ represents the number of message-passing steps. It is worth noting that when a GP has the same topology as the MLP shown in Fig. 2, it can perform arbitrary $L$ ($L \geq 2$) message-passing steps, not only 2. When $L > 2$, this can be interpreted as the output neurons not producing an output immediately upon receiving the first message but rather waiting for further ones. We also provide a neuron-level equivalent representation of equation (7), which is useful in subsequent discussions:

$$\forall l \in [0, L-1], \; \forall i \in \mathcal{H} \cup \mathcal{O},$$
$$h_i^{l+1} = b_i + \sum_{j \in \mathcal{N}} A_{ij} \widehat{x}_j^l, \; \widehat{x}_i^{l+1} = \sigma(h_i^{l+1}), \quad \text{where } \widehat{x}_i^0 = \begin{cases} x_i, \text{if } i \in \mathcal{I}, \\ 0, \text{else}. \end{cases} \text{ and } z = [h_i^L]_{i \in \mathcal{O}}. \tag{8}$$

Where $\mathcal{N} = \mathcal{I} \cup \mathcal{H} \cup \mathcal{O}$. The strength of GP lies in its ability to enable perceptrons to work under any topology. It is crucial for building self-assembling neural networks, as neurons' dynamic growth and apoptosis lead to complex connectivity patterns.

### 3.2 Growth and apoptosis: Let GP assemble itself

This section introduces how to implement a GP with full self-assembly capabilities, namely SAGP. SAGP is the first to realize a fully self-assembling neural network, capable of growing new neurons and achieving neuronal apoptosis and synaptic pruning. This functionality is realized by setting assembly rules and simulating competitive pressures, thus avoiding direct encoding of the genome.

**Initial state** The SAGP begins its development from the simplest state. GP always has a fixed number of input and output neurons, but at this stage, no hidden neuron, i.e., $|\mathcal{H}| = 0$. It also includes all possible synapses from $\mathcal{I}$ to $\mathcal{O}$, totaling $|\mathcal{I}| \times |\mathcal{O}|$.

**Neuron competition & synaptic competition** During the development of the biological brain, neuron competition [33] and synaptic competition [34] play a crucial role in acquiring cognitive and memory

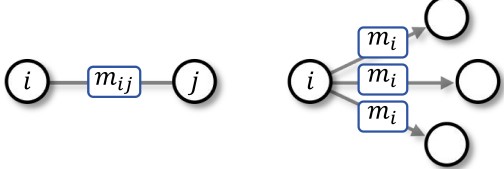

(a) Figure 4(a): Synapse-level mask (left) and neuron-level mask (right).

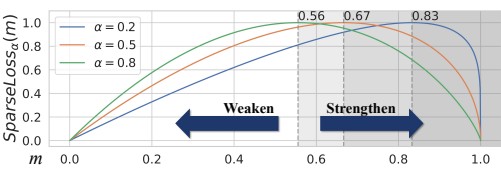

(b) Figure 4(b): The competition loss weakens most masks while only enhancing those with a competitive advantage.

abilities. We simulate this process by learnable masks with competition loss. Two types of masks are introduced in SAGP: neuron-level masks (denoted as $m_i$) and synapse-level masks (denoted as $m_{ij}$, Fig. 4a). Noting that if a hidden neuron has no outgoing synapses, it will never affect the output neurons. Therefore, the node-level mask acts on all the neuron's outgoing edges. The feature update of the GP can be rewritten as:

$$h_i^{l+1} = b_i + \sum_{j \in \mathcal{N}} m_j \cdot m_{ij} \cdot A_{ij} \widehat{x}_j^l. \tag{9}$$

All masks are learnable and can only take values of 0 or 1. It is achieved through a two-step process: first, for a learnable real number $\widehat{m} \in \mathbb{R}$, we generate a soft mask $m^{\mathrm{S}} \in [0, 1]$ by Gumbel reparameterization [35]; then, we use the Straight-Through Estimator (STE, [36]) to produce a hard mask $m = m^{\mathrm{H}} \in \{0, 1\}$ that is suitable for back-propagation:

$$m^{\mathrm{S}} = \mathsf{Gumbel\_Sigmold}(\widehat{m}), \tag{10}$$

$$m^{\mathrm{H}} = \mathsf{Stop\_Grad}(\mathbf{1}(m^{\mathrm{S}} > 0.5) - m^{\mathrm{S}}) + m^{\mathrm{S}}, \tag{11}$$

$$m = m^{\mathrm{H}}. \tag{12}$$

However, what truly enables the masks to make a competitive effect is what we refer to as the "Competition Loss":

$$\mathsf{CompLoss}_\alpha(m^{\mathrm{S}}) = C(\alpha) \cdot m^{\mathrm{S}}(1 - m^{\mathrm{S}})^\alpha. \tag{13}$$

Here, $\alpha \in (0, 1)$ is a temperature parameter, and $C(\alpha)$ is a scaling factor designed to ensure that $\mathsf{CompLoss}_a$ has a maximum value of exactly 1 over $[0, 1]$. As shown in Fig. 4b, SparseLoss weakens the less advantageous soft masks, driving them towards 0. In contrast, only the masks that dominate in competition—i.e., those with larger values—are enhanced, tending towards 1. By summing up the SparseLoss of all masks, we obtain the auxiliary loss:

$$\mathsf{AuxLoss} = \frac{\gamma_1}{|\mathcal{H}|} \sum_{i \in \mathcal{H}} \mathsf{CompLoss}_\alpha(m_i^{\mathrm{S}}) + \frac{\gamma_2}{|\mathcal{I}| + |\mathcal{H}|} \sum_{\substack{i \in \mathcal{I} \cup \mathcal{H} \\ j \in \mathcal{H} \cup \mathcal{O}}} \mathsf{CompLoss}_\alpha(m_{ij}^{\mathrm{S}}). \tag{14}$$

$\gamma_1$ and $\gamma_2$ are both weighting parameters. We achieve neuron competition and synaptic competition mechanism via back-propagation by adding AuxLoss to downstream task loss.

**Neuronal apoptosis** Biological organisms streamline the structure of their nervous systems through synaptic pruning [37] and apoptosis [9]. Similarly, when training SAGP, we introduce the following rules to remove neurons from the network to ensure the efficiency of the structure:

*Apoptosis Rule: Remove hidden neuron $i$ from set $\mathcal{H}$ and remove all synapses connected to $i$ if $m_i^{\mathrm{S}} < \beta \cdot \mathsf{Mean}_{j \in \mathcal{H}}(m_j^{\mathrm{S}}).$*

Here, $\beta \in (0, 1)$. The soft mask of a hidden neuron reflects its status in the competition. The Apoptosis Rule removes neurons that are relatively weaker within the overall population, as their influence on the outcome is negligible, thereby reducing computational overhead.

**Neuronal growth** Organisms tend to reproduce faster under reduced competitive pressure [38]. In SAGP, if no neurons have been pruned over $N$ consecutive training epochs, this may indicate that competition has sufficiently stabilized, with each neuron occupying a corresponding niche. At this point, we add new hidden neurons to increase competitive pressure:

*Growth Rule: Add a new neuron to $\mathcal{H}$ and add all possible synapses connected to this new neuron to the network if no neurons are pruned for $N$ consecutive epochs.*

As the neural network trains, the number of hidden neurons dynamically increases or decreases based on two rules. Synapses or neurons in the network may also become temporarily ineffective due to insufficient competitiveness (corresponding to a small mask value). See Appendix C.3 for implementation details.

**Assembling dynamics** We visualized the self-assembly process of SAGP (Fig. 5a and Fig. 5b) and its performance variations (Fig. 5c). The results show that the self-assembly undergoes three distinct stages: First, the network rapidly expands to near its maximum size within a short period (approximately 0 to 1000 epochs); next, intense competition occurs between neurons and synapses, with the majority of neuronal apoptosis and synaptic pruning happening during this stage (approximately 1000 to 10000 epochs); finally, the network topology stabilizes, with the number of neurons remaining almost constant while the number of synapses slowly decreases (approximately 10000 to 50000 epochs). Even as the network size decreases, its performance keeps improving, indicating that the information density in the parameters is increasing. Interestingly, this trend is similar to the changes in human cortical volume: the cortex rapidly reaches its maximum volume during childhood. It then gradually decreases in size over a long maturation period, enhancing its functionality [39].

**Hyperparametes** Several hyperparameters are introduced in SAGP, including $L$, $\alpha$, $\gamma_1$, $\gamma_2$, $\beta$, and $N$. We provide a set of empirical hyperparameters, which are consistently applied across all experiments discussed in this paper (See Appendix C.2). While hyperparameter tuning typically yields better performance for specific tasks or datasets, by fixing these parameters, we demonstrate the strong adaptability of SAGP in tackling complex environments.

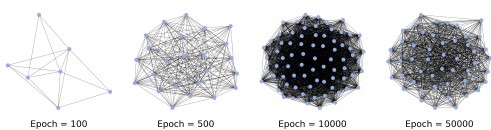

(a) Figure 5(a) We visualize the perceptron topology at several time points, where only the hidden neurons and their connections are depicted, while input and output neurons are omitted.

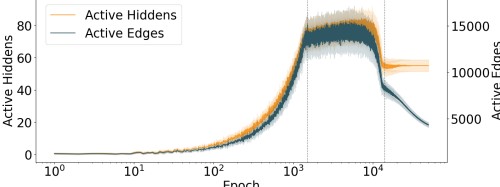

(b) Figure 5(b) Results of training SAGP on the FSDD dataset. We first present the change in the number of active neurons and synapses during training, with an apparent three-stage characteristic observed

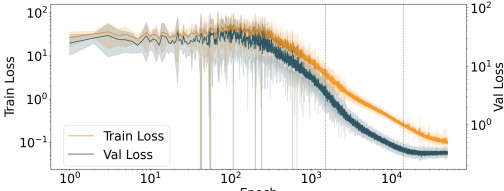

(c) Figure 5(c) Trends in the training and validation losses during the training of SAGP on the FSDD dataset.

**Complexity** Let the batch size be $B$. The computational complexity of one forward propagation in SAGP is $\mathcal{O}(BLS)$, where $L$ and $S$ represent the number of message-passing steps and the number of synapses, respectively. $S$ can be further expressed as $\lambda(n_1+n_2+n_3)^2$, where $n_1$, $n_2$, and $n_3$ represent the numbers of input, hidden, and output neurons, respectively, and $\lambda$ denotes the density of the adjacency matrix. Therefore, the complexity of SAGP is also expressed as $\mathcal{O}(\lambda BL(n_1 + n_2 + n_3)^2)$.

## 4 Experiment

**Overview** In the experiment, we answer three questions: first, the impact of topological structure on the perceptron model; second, the performance of SAGP on deep learning tasks; and third, the inspiration that self-assembling neural networks provide for modern deep learning. Three domains of the dataset are used: text, audio, and images. We also conducted experiments on deep graph models and temporal models. See Appendix C.1 for more dataset details.

### 4.1 Perceptron topology

*Question 1: Is a multi-layered connection like MLP always the best perceptron topology?*

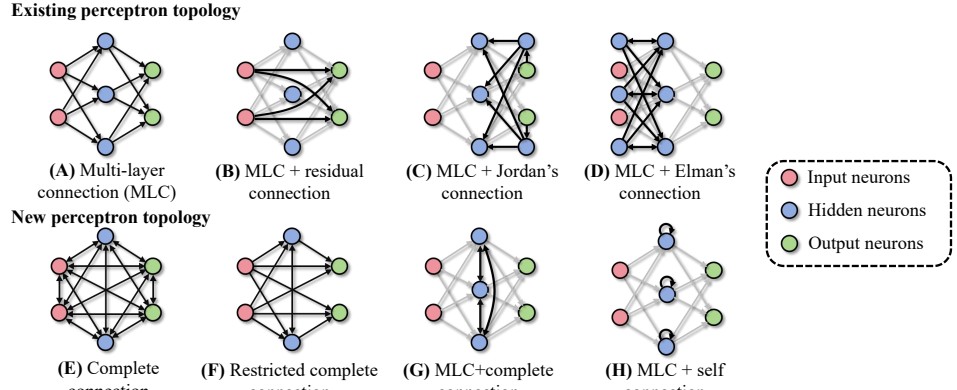

**Existing perceptron topology**

**(A)** Multi-layer connection (MLC)  **(B)** MLC + residual connection  **(C)** MLC + Jordan's connection  **(D)** MLC + Elman's connection

**New perceptron topology**

**(E)** Complete connection  **(F)** Restricted complete connection  **(G)** MLC+complete connection  **(H)** MLC + self connection

○ Input neurons
○ Hidden neurons
○ Output neurons

Figure 7: We conduct experiments on 8 different topologies of graph perceptrons to investigate the impact of neuron connectivity patterns. Topologies (A) to (D) are reported in existing literature, while (E) to (H) are new topologies designed by us.

With a fixed budget of 128 hidden neurons, we investigated the performance of 8 different topologies for graph perceptrons (Table 1). The results show that no single topology consistently outperforms others across different datasets and metrics. This suggests that mechanisms like self-assembly, which dynamically adjust the perceptron topology, have the potential to deliver better performance than fixed multi-layer topology. We select the FSDD dataset with the smallest performance gap for further investigation. We report the convergence speed of the GP (Fig. 6 left) and the impact of hidden neuron count on performance (Fig. 6 right). We found that the eight topologies can be divided into two categories: The first category includes topologies with strict hierarchical connection structures like (A), (C), (D), and (H), which converge faster but suffer from significant performance degradation at low budgets. The second category consists of topologies with non-strict hierarchical connections like (B), (E), (F), and (G), which converge slower but perform well even with a small number of hidden neurons. Thus, another potential advantage of general-topology perceptrons is that they may achieve performance comparable to multilayer-connected ones with a much smaller size. However, multilayer perceptrons have a computational efficiency advantage since they can be implemented using straightforward matrix multiplications rather than message-passing mechanisms.

| Dataset | Metric | (A) | (B) | (C) | (D) | (E) | (F) | (G) | (H) |
|---|---|---|---|---|---|---|---|---|---|
| Pho. | F1-mi. | **.877** | .870 | .865 | .866 | OOM | .876 | .861 | .873 |
|  | F1-ma. | **.843** | .836 | .828 | .831 | OOM | **.843** | .821 | .838 |
|  | AUC-mi. | .985 | .986 | .985 | .985 | OOM | **.987** | .985 | .985 |
|  | AUC-ma. | **.988** | **.988** | .987 | .987 | OOM | **.988** | .987 | .987 |
| Com. | F1-mi. | .780 | .780 | **.787** | **.787** | OOM | .780 | .779 | .784 |
|  | F1-ma. | .714 | .710 | .710 | .715 | OOM | .713 | .714 | **.720** |
|  | AUC-mi. | .972 | .971 | .973 | .971 | OOM | .971 | **.974** | .973 |
|  | AUC-ma. | **.978** | .977 | **.978** | .976 | OOM | .976 | **.978** | .976 |
| ESC. | F1-mi. | .330 | .348 | .348 | .343 | .338 | **.368** | .338 | .322 |
|  | F1-ma. | .317 | .338 | .327 | .324 | .323 | **.352** | .320 | .304 |
|  | AUC-mi. | .886 | .882 | .878 | .886 | .873 | .877 | .879 | **.892** |
|  | AUC-ma. | .880 | .875 | .871 | .877 | .867 | .873 | .873 | **.885** |
| FSD. | F1-mi. | .937 | .936 | .931 | .934 | .936 | .937 | **.938** | .933 |
|  | F1-ma. | .938 | .936 | .931 | .934 | .936 | .938 | **.939** | .933 |
|  | AUC-mi. | .995 | .996 | .995 | **.997** | **.997** | .996 | **.997** | .996 |
|  | AUC-ma. | .996 | **.997** | .996 | .996 | .996 | .996 | **.997** | .996 |
| Fas. | F1-mi. | .840 | .825 | .849 | .845 | .847 | .819 | .848 | **.855** |
|  | F1-ma. | .840 | .824 | .849 | .844 | .848 | .817 | .845 | **.854** |
|  | AUC-mi. | .989 | .984 | **.990** | .988 | .989 | .984 | .989 | **.990** |
|  | AUC-ma. | .986 | .979 | .985 | .984 | .985 | .979 | .985 | **.987** |
| CIF. | F1-mi. | **.466** | .374 | .465 | .417 | OOM | .374 | .462 | .453 |
|  | F1-ma. | **.461** | .370 | .456 | .412 | OOM | .365 | .453 | .446 |
|  | AUC-mi. | **.867** | .796 | .862 | **.867** | OOM | .795 | .864 | .860 |
|  | AUC-ma. | .865 | .796 | **.866** | .829 | OOM | .793 | .864 | .858 |

Table 1: The performance of graph perceptrons with different topologies when the number of hidden neurons is fixed 128. OOM indicates out of memory. The **best results** are highlighted.

*Answer 1: MLPs strike an excellent balance between performance and cost, but GPs can achieve the same or even higher performance with a more streamlined topology.*

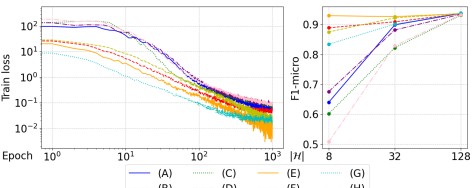

Figure 6: The graph perceptron's convergence speed (left) and its sensitivity to the number of hidden neurons (right).

| | | Scale | | Metric | | | |
|---|---|---|---|---|---|---|---|
| | | $|\mathcal{H}|$ | #Synapse | F1-micro (Accuracy) | F1-macro | AUC-micro | AUC-macro |
| Photo | MLP-b | 512 | 385536 | $.8687_{\pm.0053}$ | $.8314_{\pm.0064}$ | $.9851_{\pm.0010}$ | $.9867_{\pm.0002}$ |
| | SAGP-b | 0.20(0.04%) | 4196(1.09%) | $.8704_{\pm.0047}(100.20\%)$ | $.8354_{\pm.0058}(100.48\%)$ | $.9856_{\pm.0011}(100.05\%)$ | $.9881_{\pm.0003}(100.14\%)$ |
| | SAGP-l | 0.00(0.00%) | 4040(1.05%) | $.8671_{\pm.0015}(99.82\%)$ | $.8335_{\pm.0017}(100.25\%)$ | $.9855_{\pm.0004}(100.04\%)$ | $.9882_{\pm.0001}(100.15\%)$ |
| Computers | MLP-b | 1024 | 659968 | $.7820_{\pm.0044}$ | $.7073_{\pm.0058}$ | $.9696_{\pm.0012}$ | $.9769_{\pm.0006}$ |
| | SAGP-b | 0.00(0.00%) | 4376(0.66%) | $.7687_{\pm.0032}(98.30\%)$ | $.6986_{\pm.0045}(98.77\%)$ | $.9657_{\pm.0013}(99.60\%)$ | $.9752_{\pm.0003}(99.83\%)$ |
| | SAGP-l | 0.00(0.00%) | 3810(0.58%) | $.7677_{\pm.0040}(98.17\%)$ | $.6992_{\pm.0040}(98.85\%)$ | $.9660_{\pm.0013}(99.63\%)$ | $.9752_{\pm.0003}(99.83\%)$ |
| ESC-50 | MLP-b | 1024 | 379904 | $.3297_{\pm.0096}$ | $.2993_{\pm.0102}$ | $.8888_{\pm.0036}$ | $.8841_{\pm.0036}$ |
| | SAGP-b | 98.20(9.59%) | 20874(6.18%) | $.3190_{\pm.0126}(96.75\%)$ | $.3039_{\pm.0164}(101.54\%)$ | $.8715_{\pm.0027}(98.05\%)$ | $.8650_{\pm.0030}(97.84\%)$ |
| | SAGP-l | 106.6(10.41%) | 11251(2.96%) | $.3438_{\pm.0153}(104.28\%)$ | $.3305_{\pm.0160}(110.42\%)$ | $.8711_{\pm.0025}(98.01\%)$ | $.8659_{\pm.0027}(97.94\%)$ |
| FSDD | MLP-b | 2048 | 883712 | $.9467_{\pm.0065}$ | $.9473_{\pm.0056}$ | $.9979_{\pm.0003}$ | $.9975_{\pm.0004}$ |
| | SAGP-b | 55.10(2.69%) | 6246(0.70%) | $.9142_{\pm.0112}(96.57\%)$ | $.9149_{\pm.0111}(96.64\%)$ | $.9925_{\pm.0012}(99.46\%)$ | $.9923_{\pm.0013}(99.48\%)$ |
| | SAGP-l | 55.00(2.69%) | 4312(0.49%) | $.9120_{\pm.0142}(96.33\%)$ | $.9132_{\pm.0141}(96.40\%)$ | $.9917_{\pm.0010}(99.38\%)$ | $.9919_{\pm.0011}(99.44\%)$ |
| Fasion MNIST | MLP-l | 1024 | 668672 | $.8668_{\pm.0049}$ | $.8651_{\pm.0046}$ | $.9924_{\pm.0005}$ | $.9894_{\pm.0006}$ |
| | SAGP-l | 2.90(0.28%) | 3247(0.49%) | $.8321_{\pm.0073}(96.00\%)$ | $.8307_{\pm.0067}(96.02\%)$ | $.9869_{\pm.0009}(99.45\%)$ | $.9818_{\pm.0012}(99.23\%)$ |
| CIFAR-10 | MLP-l | 768 | 920064 | $.5201_{\pm.0027}$ | $.5168_{\pm.0037}$ | $.8975_{\pm.0013}$ | $.8941_{\pm.0011}$ |
| | SAGP-l | 65.60(8.54%) | 90674(9.86%) | $.5132_{\pm.0068}(98.67\%)$ | $.5095_{\pm.0073}(98.59\%)$ | $.8918_{\pm.0032}(99.42\%)$ | $.8880_{\pm.0035}(99.32\%)$ |

Table 2: A comprehensive comparison between SAGP and MLP. We report the network scale after training and model performance under 4 metrics. "-l" refers to the performance on the last epoch and "-b" refers to the performance on the best epoch.

## 4.2 Self-assembling graph perceptrons

*Question 2: What are the advantages of a graph perceptron with full self-assembly capacity?*

We compared two perceptron models: SAGP and MLP (Table 2). The only previous self-assembling model NDP was excluded from the comparison due to resource constraints (a single training run exceeding 12 hours on a single Nvidia RTX 4090). The suffix "-b" indicates the epoch with the *best* performance during training, i.e., the epoch that achieves the lowest loss on the validation set. The suffix "-l" refers to the *last* training epoch when convergence is reached. MLPs report results from the best epoch by default unless the dataset lacks a validation set. SAGP-b exhibits better performance, while SAGP-l has a smaller topology size. Compared to MLP, SAGP uses only $0\% \sim 10.41\%$ of hidden neurons and $0.49\% \sim 9.86\%$ of synapses, achieving performance ranging from $96\% \sim 110.42\%$. Following the setup in [7], we compared SAGP and NDP on a toy dataset, Digit (Table 3). We found that SAGP's average runtime per epoch improved by over 10,000 times. Due to neuronal apoptosis, SAGP

| | $|\mathcal{H}|$ | Average time per epoch (s) | Accuracy (%) |
|---|---|---|---|
| NDP | 48 | $1.58 \times 10^2$ | $93.0 \pm 2.9$ |
| SAGP | 0 | $1.36 \times 10^{-2}$ | $99.8 \pm 0.1$ |

Table 3: Following the same settings as in [7], we fairly compare NDP and SAGP on the toy dataset Digit [7].

| | | $|\mathcal{H}|$ | #Synapse | F1-micro |
|---|---|---|---|---|
| Pho. | GAMLP-b | 512 | 385536 | .9284 |
| | GAMLP+SAGP-b | 43.60(8.52%) | 23991(6.22%) | .9236(99.48%) |
| | GAMLP+SAGP-l | 25.50(4.98%) | 5381(1.40%) | .9164(98.71%) |
| Com. | GAMLP-b | 1024 | 659968 | .8546 |
| | GAMLP+SAGP-b | 19.30(1.88%) | 13259(2.01%) | .8620(100.9%) |
| | GAMLP+SAGP-l | 0.70(0.07%) | 526.3(0.08%) | .8527(99.78%) |
| ESC. | LSTM-b | 1024 | 379904 | .3321 |
| | LSTM+SAGP-b | 117.2(11.4%) | 28624(7.53%) | .3350(100.9%) |
| | LSTM+SAGP-l | 117.2(11.4%) | 28139(7.41%) | .3445(103.7%) |
| FSD. | LSTM-b | 2048 | 883712 | .9543 |
| | LSTM+SAGP-b | 81.4(3.97%) | 14481(1.64%) | .9285(97.30%) |
| | LSTM+SAGP-l | 68.7(3.35%) | 10652(1.21%) | .9205(96.46%) |
| Fas. | LeNet-l | 1024 | 398336 | .8979 |
| | LeNet+SAGP-l | 25.5(2.49%) | 5853(1.47%) | .8673(96.59%) |
| CIF. | LeNet-l | 768 | 438784 | .6521 |
| | LeNet+SAGP-l | 25.8(3.36%) | 26562(6.05%) | .6650(102.0%) |

Table 4: We replaced the MLP as the nonlinear transformation layers of several classic models with SAGP and evaluated its performance.

eventually removed all hidden neurons, achieving significantly better performance with only the synapses between input and output neurons. We also experimented with the potential of SAGP as a submodule in other models (Table 4). Specifically, we replaced the MLP used for generating classification outputs in GAMLP [40], LSTM [41], and LeNet [42] with SAGP. The results show that SAGP can achieve comparable performance while maintaining a significantly simplified topology. Notably, reinforcement learning-trained NDP cannot be integrated into these models.

*Answer 2: SAGP achieves performance comparable to MLP with a smaller topology, while being far more efficient and flexible than NDP.*

### 4.3 Inspiration for deep learning

*Question 3: In what fields can SAGP show its potential?*

**Model Interpretability** A significant challenge of modern deep learning models is the lack of interpretability. We visualize the out-degree of each pixel (input neuron) in the SAGP for CIFAR-10. We found that the pixels with high degrees are concentrated in the image's central region, where the image's main subject typically occupies. When selecting the top 50% of pixels by degree, the central semantics of the image are still preserved (fig. 8). This suggests that even a purely perceptron-based model like SAGP might exhibit some clues about the reasoning behind its judgments, similar to how humans respond.

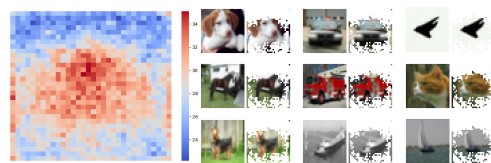

Figure 8: In the CIFAR-10 dataset, each pixel is treated as an input neuron. We visualize the out-degree of all pixels (left) in SAGP and display the image in which only the top 50% of the pixels with the largest out-degrees are retained (right).

| Backbone | Ratio | XGBoost | LassoNet | GradEnFS | Ours |
|---|---|---|---|---|---|
| GCN | 5% | $.7794_{\pm.0064}$ | $.8301_{\pm.0062}$ | $.8041_{\pm.0056}$ | $\mathbf{.8345_{\pm.0050}}$ |
| | 10% | $.7879_{\pm.0126}$ | $.8435_{\pm.0050}$ | $.8236_{\pm.0061}$ | $\mathbf{.8450_{\pm.0057}}$ |
| | 20% | $.8270_{\pm.0043}$ | $\mathbf{.8543_{\pm.0054}}$ | $.8451_{\pm.0085}$ | $.8529_{\pm.0059}$ |
| GAT | 5% | $.8137_{\pm.0094}$ | $.8473_{\pm.0078}$ | $.8233_{\pm.0070}$ | $\mathbf{.8490_{\pm.0042}}$ |
| | 10% | $.8305_{\pm.0118}$ | $.8480_{\pm.0068}$ | $.8359_{\pm.0054}$ | $\mathbf{.8517_{\pm.0047}}$ |
| | 20% | $.8407_{\pm.0037}$ | $\mathbf{.8619_{\pm.0052}}$ | $.8527_{\pm.0106}$ | $.8599_{\pm.0036}$ |

Table 5: We selected the top 5%, 10%, and 20% of features based on the out-degree as subsets, and compared them with state-of-the-art feature selection methods on the Computers dataset. The numbers in the table represent the accuracy of the semi-supervised classification task based on the selected features.

**Feature selection** Sometimes we want to determine which data preprocessing method is more effective or select the most critical subset from many features to reduce computation. These issues are collectively referred to as feature selection problems. Our experiments show that after training with SAGP, the out-degree of input features (input neurons) can indicate feature importance. In the FSDD audio dataset, we find that Mel-Frequency Cepstral Coefficients (MFCC) and Mel frequency spectrogram (Mel) features are much more effective than chroma

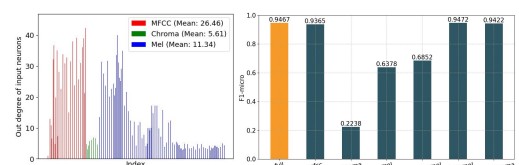

Figure 9: When training SAGP on the FSDD dataset, we present the out-degree statistics for three types of input features (left), and the results of training an MLP with different combinations of input features (right).

features (Fig. 9). On the Computers datasets, we selected the top 5%, 10%, and 20% of features based on their out-degree, and tested them with backbone models GCN [43] and GAT [44]. The classical non-deep semi-supervised feature selection method XGBoost [45], as well as the state-of-the-art deep semi-supervised feature selection methods LassoNet [46] and GradEnFS [47], were used for comparison (Table 5). The results show that SAGP significantly outperforms XGBoost and GradEnFS, and is also competitive with LassoNet, despite SAGP not being specifically designed for feature selection tasks.

*Answer 3: The out-degree of input neurons of a well-trained SAGP can be used in areas such as model interpretability and feature selection.*

## 5 Conclusion

We introduced SAGP, a graph-structured perceptron model with full self-assembly capabilities inspired by the growth process of the human brain. SAGP optimizes its topology and enhances its functionality through neuron growth, competition, and apoptosis. We highlighted the advantages of SAGP in terms of structural simplification and assembly speed and explored its potential in interpretability and feature selection.

## Acknowledgments

The research is supported by the National Key R&D Program of China (2023YFB2703700, 2022YFA1004800, 2025YFF1207900, 2025YFC3409300), and the National Natural Science Foundation of China (62176269, T2350003, T2341007, 12131020, 42450084, 42450135, 12326614, and 12426310)

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
