# OpenReview forum: "Self-Assembling Graph Perceptrons"
_NeurIPS.cc/2025/Conference — NeurIPS 2025 spotlight_

### Official Review · Reviewer_6vGG · 2025-06-10

**Clarity:** 3
**Significance:** 3
**Originality:** 3
**Rating:** 4
**Confidence:** 1

**Summary:**

The paper introduces a genuinely innovative approach combining Graph Perceptrons (GP) as a generalization of MLPs with self-assembly mechanisms inspired by biological neural development.

**Questions:**

Q1: Comparison against state-of-the-art architectures should also be provided for completeness (not only MLP). For instance, CNN results for image datasets.

Q2: How would the self-assembly principles scale to modern large-scale architectures and datasets? The computational complexity O(λBL(n₁ + n₂ + n₃)²) raises concerns about practical scalability beyond the relatively simple experimental settings presented.

**Ethical Concerns:**

["NO or VERY MINOR ethics concerns only"]

**Limitations:**

Missing limitations section in the main text.

**Quality:**

3

**Strengths And Weaknesses:**

The paper presents a novel bio-inspired self-assembly mechanism with strong theoretical foundations and impressive efficiency gains, achieving comparable performance with only 0.49%-9.86% of synapses compared to MLPs while demonstrating over 10,000x speedup compared to previous methods. The comprehensive experimental evaluation across multiple domains and integration with existing architectures shows practical potential.

---

> ### Author Rebuttal · Authors · 2025-07-30
>
> Thank you for your careful review, we now respond to your question below.
>
> # Regarding Q1
>
> We understand your concerns regarding the baselines in Table 2. However, our intended conclusion in Table 2 is that "SAGP can achieve nearly identical performance to MLP using a much sparser topology than MLP," not that "SAGP surpasses MLP in accuracy." Adding more powerful comparison methods to Table 2 would be easy, but it wouldn't add any useful conclusions. On the one hand, MLP is almost the simplest neural network, so more powerful methods (such as CNNs) clearly achieve superior performance and are therefore likely to outperform SAGP. On the other hand, these methods include not only the MLP as a module but also many other modules, such as convolutional layers and regularization layers. It's difficult to say how many "neurons" or "synapses" a convolutional layer or regularization layer has. We typically use these terms only in relation to MLPs, so including other models in Table 1 would obscure the comparison of topology size.
>
> We specifically explore other applications of SAGP besides reducing topology size in Section 5.3 of the main paper, such as feature selection. At this point, we compare SAGP with the most advanced methods in this field and find that SAGP remains highly competitive.
>
>
>
> # Regarding Q2
>
> In fact, the computational complexity of SAGP ($O(\lambda BL(n_1+n_2+n_3)^2)$) is equivalent to that of MLP in some sense. We consider a two-layer MLP and ignore the activation function and bias (because they are not the main factors affecting the complexity), which can be expressed as:
>
> $$
> \boldsymbol{z}=\boldsymbol{W}^T_2\boldsymbol{W}^T_1\boldsymbol{x}
> $$
>
> Where $\boldsymbol{x} \in \mathbb R^{|\mathcal I|}$, $\boldsymbol{W}_1^T \in \mathbb R^{|\mathcal H| \times |\mathcal I|}$,$\boldsymbol{W}_2^T \in \mathbb R^{|\mathcal O| \times |\mathcal H|}$ and $\boldsymbol{z} \in \mathbb R^{|\mathcal O|}$
> .
>
> Recall that the time complexity of multiplying a $(n\times m)$ matrix by a $(m \times p)$matrix is $\Theta(nmp)$ and the space complexity is $\Theta(1)$. Therefore, the time and space complexities of computing $z$ are $\Theta(|\mathcal H|\times|\mathcal O| + |\mathcal H|\times|\mathcal I|)$ and $\Theta(1)$, respectively. Note that the total number of synapses $S_{mlp}$ in this MLP is also $|\mathcal H|\times|\mathcal O| + |\mathcal H|\times|\mathcal I|$, so the time complexity can also be expressed as $\Theta(S_{mlp})$.
>
> A simplified GP can be expressed as:
>
> $$
> \boldsymbol{z} = \left((\boldsymbol{A}^T)^2\begin{bmatrix}\boldsymbol{x} \\ 0_{|\mathcal H| + |\mathcal O|}\end{bmatrix}^T\right)_{[-|\mathcal O|:]}
> $$
>
> The definition of $\boldsymbol A$ is consistent with that of Formula 6 in the paper. GP use sparse matrix multiplication in its implementation, which in **once sparse matrix multiplication**, the time complexity and space complexity are both $\Theta(S_{gp})$ , Here, $S_{gp}$ represents the number of synapses in GP, which is equivalent to the number of non-zero elements in $\boldsymbol A$. So for a GP with two message-passing, the time complexity and space complexity are $\Theta(2S_{gp})$ and $\Theta(S_{gp})$ respectively.
>
> The above conclusions can be extended to any layer of $L$-layer MLP or any $L$-times message-passing GP. Now, we Let the batch size is $B$, and the MLP and GP training rounds are $k_{mlp}$ and $k_{gp}$ respectively, then the total training complexity comparison is:
>
> |      | Time                        | Space               |
> | ---- | --------------------------- | ------------------- |
> | MLP  | $$\Theta(k_{mlp}BS_{mlp})$$ | $$\Theta(B)$$       |
> | GP   | $$\Theta(k_{gp}BLS_{gp})$$  | $$\Theta(BS_{gp})$$ |
>
> $\lambda(n_1+n_2+n_3)^2$ is actually the expansion of $S_{gp}$.
>
> **Here, we would like to ask you to pay special attention to one detail**: in the table, it seems that the time complexity of GP is one factor $L$ higher than that of MLP, but in fact, when $\Theta(k_{mlp})=\Theta(k_{gp})$, the complexity of the two is equivalent. This is because $S_{mlp}$ increases linearly with $L$, but $S_{mlp}$ is constant with $L$. In other words, $\Theta(S_{mlp})=\Theta(LS_{gp})$.
>
> For the complexity of SAGP, we only need to replace $S_{gp}$ with $\max S_{sagp}$.
>
> In particular, in experimental observations, we found that:
>
> |                            | Photo  | Computers | ESC-50 | FSDD   | Fashion-MNIST | CIFAR-10 |
> | -------------------------- | ------ | --------- | ------ | ------ | ------------- | -------- |
> | $S_{mlp}$                  | 385536 | 659968    | 379904 | 883712 | 668672        | 920064   |
> | $$L \times \max S_{sagp}$$ | 30126  | 31257     | 66519  | 47565  | 184572        | 625923   |
>
> This further illustrates that $\Theta(S_{mlp})=\Theta(L\max S_{sagp})$. Therefore, at least in theory, the time cost of a single round of inference for SAGP is no more than that of MLP. But in practice, SAGP does cost more than MLP (we clearly acknowledge this in the "Limitations" section of the Appendix). Based on the complexity comparison table above, we can reduce the cost of SAGP in two ways: reducing $k_{gp}$ or reducing $\max S_{sagp}$. Here, we offer two strategies to achieve this.
>
> 1. **Initializing SAGP using MLP.** Although we describe SAGP as developing from a minimal topology in the main text, please note that the neuron/synapse growth and apoptosis process in SAGP is actually independent of initialization. Therefore, we can first train an MLP on the same dataset and then use it as the initialization for SAGP. This effectively reduces the optimization difficulty and the number of required epochs.
> 2. **Limiting the in-degree of neurons.** Thanks to the reviewer **4y96** for the great suggestion. We note that directly limiting the maximum degree of neurons is a simple and reasonable approach - after all, it is unlikely that a neuron in a biological brain is connected to all other neurons.
>
> We verified these two ideas on the Photo dataset as follows:
>
> |          | Epoch($k_{sagp}$) | Training Time | Max Memory     | ACC         | #Synapse  |
> | -------- | ----------------- | ------------- | -------------- | ----------- | --------- |
> | SAGP     | 7150              | 267s          | 7020Mb         | .8687       | 5316      |
> | SAGP+I   | 5681 (-21%)       | 220s (-17%)   | 10633Mb (+15%) | .8549 (-2%) | 4871(-8%) |
> | SAGP+L   | 6977 (-2%)        | 251s (-6%)    | 6459Mb (-8%)   | .8574 (-1%) | 5301(-0%) |
> | SAGP+I+L | 6430 (-10%)       | 231s (-13%)   | 6391Mb (-9%)   | .8594 (-1%) | 5514(+4%) |
>
> Where SAGP+I means using MLP as initialization. SAGP+L means limiting in-degree. SAGP+I+L means applying both at the same time. When +I is applied, we first construct and train a three-layer MLP with 50% of the budgeted hidden neurons as the initialization of SAGP, and then use the remaining 50% to grow according to the rules of SAGP. When “+L” is applied, only the top 100 weighted incoming edges in each neuron are used for message-passing. We use the number of neurons remaining unchanged in 100 epochs as the stopping condition.
>
> We found that using the "Initializing SAGP using MLP" and "Limiting the in-degree of neurons" strategies played a positive role in optimizing the time and space complexity of SAGP, respectively, and the two strategies can work simultaneously while keeping the accuracy (ACC) and network size (#Synapse) approximately unchanged. Given that we have only conducted preliminary explorations of these two strategies, we believe that after further detailed design and optimization, it is entirely possible to apply the self-assembly mechanism to larger models and datasets.

---

> > ### Comment · Area_Chair_m4xC · 2025-08-07
> >
> > Dear Reviewer 6vGG,
> >
> > Thank you for your previous input. As the discussion deadline is approaching, we kindly remind you to review the author's response and continue the discussion if you have any further comments or questions.
> >
> > Your timely feedback will be greatly appreciated to ensure a thorough review process.
> >
> > Thank you very much for your attention!
> >
> > Best regards,
> >
> > AC

---

### Official Review · Reviewer_rmb6 · 2025-06-25

**Clarity:** 4
**Significance:** 3
**Originality:** 4
**Rating:** 6
**Confidence:** 4

**Summary:**

Inspired by the function of the human brain, this paper introduces the graph perceptron (GP) together with an assembling mechanism which allows for neuron growth and apoptosis. The authors investigate the training process of the GP, finding similarities to how the human brain functions. In competition with standard MLPs, the GP obtains similar performance with a small fraction of the MLP's size. Additionally, the GPs abilities for feature selection is studied, finding it to match or outperform state of the feature selection algorithms.

**Questions:**

Major questions/suggestions:
1. As explained in the introduction, the GP architecture is inspired by the human brain, and the authors find the training dynamics to be similar to that of the human brain (ll. 210 - 214). This suggests to me that it might be possible to explain some aspects of the human brain with this model, extending the impact of this model to the neuroscience community (although I understand that this is not the scope of the current paper). This seems like a missed opportunity to me, adding the authors' thoughts on this to the discussion would be interesting.
2. (similar to weakness 1) For the experiments in table 2, being the performance comparison between MLP and the GP, additionally stating the runtime and maximum memory would be interesting and relevant for practitioners.
3. (figure 6 left plot) I find it interesting that the different models seemingly have very different train loss, even after one epoch, being almost at random initialization, for which I would expect similar loss for all models. Adding the authors' thoughts to this would be valuable.
4. The appendix offers a lot of helpful information, it would be useful to point to this at relevant points in the manuscript.


Minor questions/suggestions:
1. What is the toy dataset Digit used for table 3? I can not find an explanation.
2. What do the numbers stated in table 5 represent?
3. l. 142 defines $\mathcal{N}$, however this is used neither in the text nor in eq.8?
4. After reading only question 3 and answer 3 they do not fit well to each other. After reading the main text the authors' thoughts become clear, but I would suggest to reformulate answer 3 such that it is understandable withou fully reading the main text.
5. l.352 typo slection
6. figure 5c: using a single y-axis would increase the readability of the plot, at first glance it looks as if the train loss is higher than the validation loss
7. Pointing to the dataset explanation in the appendix directly in the caption of table 1 would make it easier to find.

**Ethical Concerns:**

["NO or VERY MINOR ethics concerns only"]

**Final Justification:**

This is a strong paper with clear presentation and many helpful figures, making it a pleasure to read. It is easy to follow the authors' thoughts even when coming from a different field. Additionally, the proposed model is rigorously investigated with thoughtful experiments, which moreover demonstrate the superior performance of the proposed model.

Therefore I strongly recommend accepting this paper.

**Limitations:**

I do not see any additional limitations to those the authors already discuss in the appendix. However, I would encourage the authors to make these more prominent in the main text, at least by pointing to the appendix in the conclusion.

**Paper Formatting Concerns:**

I do not have any paper formatting concerns.

**Quality:**

4

**Strengths And Weaknesses:**

Strengths:
1. The graph perceptron is well motivated by the functioning of the human brain. During the training the authors find similarities to how humans learn, without building this explicitly into the model. This gives the model relevance beyond its application in machine learning tasks.
2. The graph perceptron is thoroughly studied over multiple aspects of its behaviour (training dynamics, performance comparison against MLP and state of the art, feature selection), the experiments are well designed, performed and presented.
3. Matching the MLP performance with only 5% of its size is an impressive result.
4. The paper is a pleasure to read. The authors' thoughts are explained well by both text and figures.

Weaknesses:
1. For the experiments only the size of the final GP in comparison to the MLP is given. For practiticioners the runtime as well as maximum used memory would be as relevent. (This is discussed shortly in the appendix, but nowhere pointed to in the main text as far as I see)

---

> ### Author Rebuttal · Authors · 2025-07-30
>
> We deeply appreciate your hard work in providing a professional and thorough review. Below is our response.
>
> # Responses to weaknesses:
>
> + [W1]:
>   We provide the statistics of runtime costs that you are concerned about in the following two parts, along with a detailed comparison with MLP:
>
>  ## Training cost (Table A)
>
> |                             | Photo         | Computers     | ESC-50        | FSDD          | Fashion-MNIST(batch_size=128) | CIFAR-10 (batch_size=128) |
> | --------------------------- | ------------- | ------------- | ------------- | ------------- | ----------------------------- | ------------------------- |
> | Training epoch (MLP)        | 1000          | 1000          | 1000          | 1000          | 1000                          | 1000                      |
> | Training epoch (SAGP)       | 50000         | 50000         | 50000         | 50000         | 100000                        | 100000                    |
> | Ratio                       | $\times$50    | $\times$50    | $\times$50    | $\times$50    | $\times$100                   | $\times$100               |
> | Training Time (MLP)         | 11s           | 16s           | 12s           | 13s           | 39s                           | 43s                       |
> | Training Time (SAGP)        | 754s          | 748s          | 634s          | 674s          | 3705s                         | 4031s                     |
> | Ratio                       | $\times$68.54 | $\times$46.75 | $\times$52.87 | $\times$56.24 | $\times$95.00                 | $\times$93.74             |
> | Training Peak memory (MLP)  | 734 Mb        | 636 Mb        | 651 Mb        | 898 Mb        | 512 Mb                        | 510 Mb                    |
> | Training Peak memory (SAGP) | 7020 Mb       | 14610 Mb      | 4994Mb        | 7833Mb        | 14189 Mb                      | 15530 Mb                  |
> | Ratio                       | $\times$9.56  | $\times$22.97 | $\times$7.67  | $\times$8.27  | $\times$27.71                 | $\times$30.45             |
>
> ## Inference cost (Table B)
>
>
> |                             | Photo        | Computers    | ESC-50       | FSDD         | Fashion-MNIST(batch_size=128) | CIFAR-10 (batch_size=128) |
> | --------------------------- | ------------ | ------------ | ------------ | ------------ | ----------------------------- | ------------------------- |
> | Training Time (MLP)         | 0.03s        | 0.09s        | 0.03s        | 0.02s        | 1.23s                         | 1.98s                     |
> | Training Time (SAGP)        | 0.11s        | 0.16s        | 0.09s        | 0.08s        | 1.29s                         | 2.53s                     |
> | Ratio                       | $\times$3.66 | $\times$1.78 | $\times$3.00 | $\times$4.00 | $\times$1.04                  | $\times$1.28              |
> | Training Peak memory (MLP)  | 90 Mb        | 156 Mb       | 132 Mb       | 322 Mb       | 30 Mb                         | 78 Mb                     |
> | Training Peak memory (SAGP) | 291 Mb       | 401 Mb       | 821Mb        | 768Mb        | 39 Mb                         | 186 Mb                    |
> | Ratio                       | $\times$3.23 | $\times$2.57 | $\times$6.22 | $\times$2.38 | $\times$1.30                  | $\times$2.38              |
>
> As shown in Table A, SAGP takes roughly the same amount of time to train per round as MLP, but the greater number of rounds results in a longer training time. Furthermore, SAGP training memory overhead is approximately 20 times higher. In terms of inference cost, both time and space consumption are roughly 1-6 times that of MLP.
>
> As we mentioned in the limiation section of the appendix, we clearly understand that improving the efficiency of SAGP is a key area of future research. However, SAGP is already over 10,000 times faster than NDP, the previously considered sole self-assembling network. Furthermore, self-assembly also brings several unique benefits to the model. For example, as we mentioned in the main text, when used for feature selection, SAGP can reach or even surpass the state-of-the-art methods.
>
> Finally, our response to reviewer **4y96** discusses in detail the efficiency improvements to SAGP. If you are interested in this topic, you are welcome to read them further. Due to space limitations, we will not elaborate on these details here. We sincerely hope you will understand.
>
> # Responses to major questions/suggestions:
>
> + [Major Q1]: Explaining the behavior of biological brains with (self-assembling) neural networks is indeed a very interesting question. We would like to share some naive opinions on this. Why is the increase in cortical volume in biological brains mainly concentrated in the early stages of biological development, rather than evenly distributed throughout the whole life cycle? This question has been extensively studied in biology. But if we only explain it from the experimental results of SAGP, we think it is very similar to the ``coordinate descent method'' in optimization. Coordinate descent is a strategy whereby for a multi-parameter optimization problem, we select only one variable at a time for optimization, while keeping the other variables unchanged. MLP only optimizes the weights between neurons, while SAGP needs to optimize weights and topology. In the early stages of network optimization, since all weights are randomly initialized, there are no absolutely dominant neurons at this time. Therefore, the growth rule will play a major rule, causing the network size to expand rapidly. After a period of training, dominant neurons begin to appear, and at this time the apoptosis rule becomes dominant. The network size gradually decreases. It is as if the model first focuses on optimizing the topological structure and then focuses on optimizing the weights. We speculate that organisms have somehow learned this "coordinate descent" strategy in long-term evolution, and this strategy may be more stable for the development of a single organism than strategies that optimize both topology and weights at the same time (such as traditional random gradient descent).
>
> + [Major Q2]: Pleas see our response to Weaknesses 1 on this issue.
>
> + [Major Q3]: To explain this problem, we first repeat formula (5) in the main text, which shows how a GP after two massage-passing obtains its logits:
>
>   $$
>   \mathcal z = (A^T\sigma(A^T\begin{bmatrix}x \\ 0_{|\mathcal H| + |\mathcal O|}\end{bmatrix}^T+b)+b)_{[-|\mathcal O|:]}
>   $$
>
>   Where $|\mathcal H|$ and $|\mathcal O|$ represent the number of hidden neurons and output neurons, respectively. To simplify the situation, we now remove all activation functions and biases from the above formula:
>   $$
>   \mathcal z = \left((A^T)^2\begin{bmatrix}x \\ 0_{|\mathcal H| + |\mathcal O|}\end{bmatrix}^T\right)_{[-|\mathcal O|:]}
>   $$
>   If all elements in $A$ are randomly initialized, then the loss calculated using $z$ and the label should indeed have the same expectation. However, not all elements in $A$ are learnable. In fact, in a topology, if two neurons are connected, then the weights between them need to be randomly initialized (and learned). Otherwise, the weights are fixed to zero. Therefore, for different topologies, the non-zero positions are also different, so the obtained $z$ will also be very different, resulting in different initial losses. However, it must be warned that this is only a very simplified analysis. We ignore the activation function, bias, and do not consider other parts of the specific implementation (such as normalization). Therefore, it is difficult to have an exact relationship here that allows us to get its initial loss from the topology of the GP.
>
> + [Major Q4]: Thank you for your suggestion, which is very important for us to further improve the clarity of the paper. Now we have mentioned everything related to the experiments in the appendix in more detail in Section 4 (Experiments) in the main text: including dataset description, hyperparameters, pseudocode, ablation experiments, parameter experiments, etc. At the same time, we mentioned the section ``Differences between self-assembling networks and other topics'' in the appendix at the end of Section 2 (Related Work) in the main text. Finally, we also mentioned the limitations of SAGP (Appendix Section 5.1) in Section 5 (Conclusion) in the main text to help readers understand the model more comprehensively and objectively.
>
> # Responses to minor questions/suggestions:
>
> + [Minor Q1]: The toy dataset Digit is the dataset used in the previous self-assembling neural network NDP [7] (this dataset is actually an 8*8 pixel version of the famous MNIST dataset). We have now added additional explanations and references in the Caption of table 3.
> + [Minor Q2]: The data in Table 5 represents the accuracy of the selected feature subsets in the downstream classification task after using different feature selection methods/ ratios. The higher the number, the better the method can select the most important feature components. We have now added additional instructions in the table caption.
> + [Minor Q3]: $\mathcal N$ is explicitly used as the subscript of the summation sign in Equation 8 and Equation 9.
> + [Minor Q4]: This is a great suggestion. We have now modified Question 3 to “In what fields can SAGP show its potential ?” and Answer 3 to “The out-degree of input neurons of a well-trained SAGP can be used in areas such as model interpretability and feature selection.” to ensure that the question-answer pairs are more clearly aligned.
> + [Minor Q5]: We have fixed this typo in the latest version.
> + [Minor Q6]: In Figure 5C, since the magnitudes of the two curves are not exactly the same, using a single y-axis will make one of the trends unclear. If space permits, we will consider replacing it with two independent images in the final version.

---

> > ### Comment · Reviewer_rmb6 · 2025-08-04
> >
> > I thank the authors for their detailed and clear answer, especially appreciated are the training and inference cost tables. Adding these together with the authors' thoughts in their answer to reviewer 4y96 on how to reduce the complexity further strengthens the manuscript and reinforces my opinion that this paper should be accepted. I do not have any follow up questions.

---

### Official Review · Reviewer_4y96 · 2025-07-01

**Clarity:** 4
**Significance:** 3
**Originality:** 4
**Rating:** 6
**Confidence:** 4

**Summary:**

Inspired by flexible and evolving structures in biological neural network, the authors propose a novel perceptron model, SAGP. Proposed architecture GP generalizes perceptron models, and during training, SAGP optimizes proposed loss function to learn perceptron structure. In multiple classification datasets, SAGP achieves similar accuracies to those achieved by MLP, but with much sparser neural connections. Authors further demonstrate SAGP is useful for interpretability and feature selection.

**Questions:**

See weaknesses.

**Ethical Concerns:**

["NO or VERY MINOR ethics concerns only"]

**Final Justification:**

This is a strong paper with clear presentation, rigor, originality, and remarkable performance.

Concerns have been raised primarily about the method's complexity. However, the authors satisfactorily resolved them with theoretical proofs, empirical demonstration, and an updated version of the method that reduces complexity.

In my view, the only small weakness of the paper is the use of only classification benchmarks to evaluate the proposed method's performance. This weakness is understandable, given the other advantages that the proposed method has.

**Limitations:**

Yes.

**Quality:**

4

**Strengths And Weaknesses:**

The **key strengths** include:

- [S1 originality]. The proposed method SAGP is original. The choice to generalize perceptron models is innovative, and the architecture design is simple and neat.
- [S2 comprehensive exp.]. Analysis and experiments were comprehensive. Some results are reported in Appendix to demonstrate the consistency of the findings across different datasets. Feature selection and interpretability experiments further demonstrate usefulness of the proposed method.
- [S3 theoretical property]. Proving that MLP is a special case of GP is important (Theorem 1). Besides, mathematical derivation of GP from MLP is creative.
- [S4 clear presentation]. The presentation is highly effective. Figures clearly convey their messages. Mathematics are well explained. The argument structure in Experiment section is clearly guided.

These strengths clearly set up the paper for a top-tier conference. However, I lean toward only borderline accept since SAGP seems to have **very low applicability with two primary limitations**.

- [W1 complexity]. While SAGP’s ‘inference’ speed and memory may be comparable to those in MLP due to learned sparsity (Appendix Sec 5.1), they are substantially larger for model training. Since the complexity is square of input, hidden, and output neurons, SAGP cannot scale to larger dimensions. Indeed, SAGP reportedly becomes very dense during training (Fig. 5a), which would limit its application.
- [W2 accuracy]. Since the reported SAGP accuracies are comparable to those of MLP, they do not justify SAGP’s complexity handicap. Thus, as a user, the present work does not convincingly demonstrate empirical advantages of using SAGP.

I am willing to raise my score to accept, if either W1 or W2 is somewhat resolved during rebuttal. In light of that, I would like to make few **suggestions for future reference**.

- [suggestion 1 - faster variant of SAGP]. Have the authors considered designing a faster variant of SAGP to mitigate the complexity issue? Naively, I would think restricting synapse space may significantly reduce complexity.
- [suggestion 2 - self-supervision tasks]. Have the authors tested SAGP for self-supervision tasks? Current performance metrics are all based on classification. Since SAGP generalizes MLP, I speculate that the autoencoder with SAGP encoder and decoder may substantially outperform an autoencoder with MLPs. Even if SAGP does not outperform, given the importance of self-supervision tasks in the deep learning today, testing SAGP for self-supervision may improve the present work.

---

> ### Author Rebuttal · Authors · 2025-07-29
>
> We are grateful for your professional comments and recognition of our work. We would like to respond respectfully to the weaknesses you pointed out as follows.
>
> # 1. [Weakness1: complexity]
> To ensure a proper understanding of this weakness, we would like to reanalyze the theoretical complexity of MLP and SAGP before formally answering it.
>
> For simplicity, we ignore the activation function and bias in MLP/SAGP because they are not the main factors affecting time/memory cost. At this point, a two-layer MLP can be expressed as:
>
> $$
> \boldsymbol{z}=\boldsymbol{W}^T_2\boldsymbol{W}^T_1\boldsymbol{x}
> $$
>
> Where $\boldsymbol{x} \in \mathbb R^{|\mathcal I|}$,  $\boldsymbol{W}_1^T \in \mathbb R^{|\mathcal H| \times |\mathcal I|}$,$\boldsymbol{W}_2^T \in \mathbb R^{|\mathcal O| \times |\mathcal H|}$ and $\boldsymbol{z} \in \mathbb R^{|\mathcal O|}$。Recall that the time complexity of multiplying a $(n\times m)$ matrix by a $(m \times p)$matrix is $\Theta(nmp)$ and the space complexity is $\Theta(1)$. Now, we multiply $\boldsymbol{W}_1^T$ by $\boldsymbol{x}$, and then multiply the result by $\boldsymbol{W}_2^T$.
>
> After these two steps, the time complexity is $\Theta(|\mathcal H|\times|\mathcal O| + |\mathcal H|\times|\mathcal I|)$, and the space complexity is $\Theta(1)$ (We only consider the overhead beyond input and output). Note that the total number of synapses $S_{mlp}$ in this MLP is also $|\mathcal H|\times|\mathcal O| + |\mathcal H|\times|\mathcal I|$, so the time complexity can also be expressed as $\Theta(S_{mlp})$.
>
> Then we consider GP. For a GP that has undergone two message-passing, it can be simplified as:
>
> $$
> \boldsymbol{z} = \left((\boldsymbol{A}^T)^2[ \boldsymbol{x} \\ 0_{|\mathcal H| + |\mathcal O|}]^T  \right)_{[-|\mathcal O|:]}
> $$
>
>
>
> The definition of $\boldsymbol A$ is consistent with that of Formula 6 in the paper. GP use sparse matrix multiplication in its implementation, which in **once sparse matrix multiplication**, the time complexity and space complexity are both $\Theta(S)$, where $S$ is the number of non-zero elements in $\boldsymbol{A}$. Note that the number of non-zero elements of $\boldsymbol{A}$ is also equal to the number of synapses in the GP ($S_{gp}$), so for a GP with two message-passings, the time complexity and space complexity are $\Theta(2S_{gp})$ and $\Theta(S_{gp})$ respectively.
>
> The above conclusions can be extended to any layer of $L$-layer MLP or any $L$-times message-passing GP. Now, we Let the batch size is $B$, and the MLP and GP training epochs are $k_{mlp}$ and $k_{gp}$ respectively, then the total training complexity comparison is:
>
> |      | Time                        | Space               |
> | ---- | --------------------------- | ------------------- |
> | MLP  | $$\Theta(k_{mlp}BS_{mlp})$$ | $$\Theta(B)$$       |
> | GP   | $$\Theta(k_{gp}BLS_{gp})$$  | $$\Theta(BS_{gp})$$ |
>
> **Here, we would like to ask you to pay special attention to one detail**: in the table, it seems that the time complexity of GP is one factor $L$ higher than that of MLP, but in fact, when $\Theta(k_{mlp})=\Theta(k_{gp})$, the complexity of the two is equivalent. This is because $S_{mlp}$ increases linearly with $L$, but $S_{gp}$ is constant with $L$. In other words, $\Theta(S_{mlp})=\Theta(LS_{gp})$.
>
> For the complexity of SAGP, we only need to replace $S_{gp}$ in SP with $\max S_{sagp}$.
>
> We then give the following observation in our experiment:
>
> |                   | Photo  | Computers | ESC-50 | FSDD   | Fashion-MNIST | CIFAR-10 |
> | ----------------- | ------ | --------- | ------ | ------ | ------------- | -------- |
> | $S_{mlp}$         | 385536 | 659968    | 379904 | 883712 | 668672        | 920064   |
> | $$L\times\max S_{sagp}$$ | 30126  | 31257     | 66519  | 47565  | 184572        | 625923   |
>
> Based on the above analysis, we believe that, **theoretically, the (average) single-epoch training time complexity of SAGP is consistent with that of MLP. However, in practice, SAGP's higher time cost stems from the use of more epochs, while its higher space cost stems from the use of sparse matrix multiplication, which increases linearly with $S_{gp}$.**
>
> Next, we discuss how to improve SAGP from the two aspects of reducing time and space complexity.
>
> ## 1.1 Improve the time complexity of SAGP
>
> ### 1.1.1 Reducing the training epochs
> The key challenge in improving time complexity is reducing the number of training epochs. As shown in the results presented in Section 3.2, “Assembly Dynamics,” the number of hidden neurons and validation loss of SAGP on the FSDD dataset remain nearly constant around epoch 10,000. At this point, the ratio of the number of synapses in SAGP to the number of synapses in the MLP is 0.96%. It took an additional 40,000 epochs to reduce this ratio to 0.49%. This is actually a trade-off that depends on user preference: if users are willing to spend more time, they can obtain a smaller model. Therefore, if you want to reduce training time, directly reducing the number of training epochs is the simplest approach, which generally does not affect accuracy.
>
> ### 1.1.2 Initializing SAGP using MLP
> Although we describe SAGP as developing from a minimal topology in the main text, please note that the neuron/synapse growth and apoptosis process in SAGP is actually independent of initialization. Therefore, we can first train an MLP on the same dataset and then use it as the initialization for SAGP. This effectively reduces the optimization difficulty and the number of required epochs.
>
> ## 1.2 Improve the space complexity of SAGP
>
> ### 1.2.1 Limiting the in-degree of neurons.
> The key to improving the space complexity of SAGP is to reduce $\max S_{sagp}$. As you mentioned in [suggestion 1], limiting the synaptic space is a great approach. From a biological perspective, it is obviously unlikely that a neuron would need to receive input from all the other neurons in the brain. From a computational complexity perspective, if we limit the maximum in-degree of a neuron to $K$, the relationship between $\max S_{sagp}$ and the number of neurons will decrease from quadratic to linear.
>
> ## 1.3 Additional experimental results
> We use the number of neurons remaining unchanged in 100 epochs as the stopping condition and report the computational cost and accuracy of the two methods mentioned in 1.1.2 and 1.2.1. Among them, SAGP+I means using MLP as initialization. SAGP+L means limiting in-degree. SAGP+I+L means applying both at the same time. When +I is applied, we first construct and train a three-layer MLP with 50% of the budgeted hidden neurons as the initialization of SAGP, and then use the remaining 50% to grow according to the rules of SAGP. When “+L” is applied, only the top 100 weighted incoming edges in each neuron are used for message-passing.
>
> We apologize for the time and character limitations, so we only report the results on the dataset Pohto as follows:
>
> |          | Epoch       | Training Time | Max Memory     | ACC         | #Synapse  |
> | -------- | ----------- | ------------- | -------------- | ----------- | --------- |
> | SAGP     | 7150        | 267s          | 7020Mb         | .8687       | 5316      |
> | SAGP+I   | 5681 (-21%) | 220s (-17%)   | 10633Mb (+15%) | .8549 (-2%) | 4871(-8%) |
> | SAGP+L   | 6977 (-2%)  | 251s (-6%)    | 6459Mb (-8%)   | .8574 (-1%) | 5301(-0%) |
> | SAGP+I+L | 6430 (-10%) | 231s (-13%)   | 6391Mb (-9%)   | .8594 (-1%) | 5514(+4%) |
>
> We found that using the "MLP initialization" and "limiting in-degree" strategies played a positive role in optimizing the time and space complexity of SAGP, respectively, and the two strategies can work simultaneously while keeping the accuracy (ACC) and network size (#Synapse) approximately unchanged. Given that we have only conducted preliminary explorations of these two strategies, we believe they could perform even better with more thoughtful design.
>
> # 2. [W2 accuracy]
>
> We fully understand your concerns about the applicability of SAGP. We acknowledge that SAGP still has some gaps in efficiency compared to MLP, which is almost the simplest neural network. However, SAGP is already over 10,000 times faster than the NDP[7], which was previously the only self-assembling neural network. On the other hand, self-assembly also gives SAGP unique advantages that MLP does not have. For example, as we mentioned in Section 4.3 of the main text, we found that when SAGP is used for feature selection, it can achieve comparable or even better performance than the state-of-the-art models.This means we can leverage SAGP to accelerate the training of all other models on the same dataset.
>
> Furthermore, we have found that SAGP is particularly well-suited for **continuous learning**. We conduct an additional experiment to verify this. First, we split the 10 classes in the Fashion-Mnist dataset into two parts. We initially trained SAGP on data from only the first five classes. After the topology stabilized, we restarted the self-assembly process on data from the last five classes. At this point, we get the following results:
>
> |                           | MLP            | SAGP            |
> | ------------------------- | -------------- | --------------- |
> | ACC on first five classes | 90.7% -> 43.6% | 89.2% -> %61.7% |
>
> We found that SAGP forgets much slower than MLP on the first five classes. We hypothesize that this is due to the unique mechanism by which SAGP can learn new knowledge by generating new neuron. This may also constitute one of the unique advantages of SAGP.
>
> Finally, we apologize that due to time constraints, we were unable to report results on the self-supervised task, but this is indeed a promising idea because SAGP may learn task-specific structural information better than MLP. We will include this experiment in the future improvements of the paper.

---

> > ### Comment · Reviewer_4y96 · 2025-08-04
> >
> > Dear authors,
> >
> > I appreciate the thorough rebuttal. I have carefully reviewed it, and here are my concluding thoughts.
> >
> > **On [W1 complexity].**
> >
> > - My concern is well addressed. The authors clarified some misunderstanding I had about the complexity of the proposed method. I am surprised to find that the proposed method, despite its sophisticated design, has similar theoretical complexity with MLP. I also appreciate the efforts to improve the complexity of the proposed method.
> > - I think part of my misunderstanding stemmed from Fig. 5(a). It appears as if SAGP’s synapses would substantially outnumber those in MLP. Please consider improving the visualization to help clarify SAGP’s complexity.
> >
> > **On [W2 accuracy]**
> >
> > - My concern is somewhat addressed. I understand that the proposed method is 10,000 times faster than NDP. Also, I appreciate the updated, preliminary experimental result. It is quite interesting.
> > - However, further empirical evaluations would have been helpful (though I understand the time constraint).
> >
> > Furthermore, during rebuttal, I also checked the paper’s Appendix. The authors implemented rigorous experimental setting and reported all necessary details, which is another strength of the paper.
> >
> > Overall, this is a strong paper with rigor, originality, and remarkable performance. I recommend acceptance. For camera ready, I encourage the authors to expand on predictive performance evaluation.
> >
> > Sincerely,

---

### Official Review · Reviewer_v1ys · 2025-07-03

**Clarity:** 3
**Significance:** 2
**Originality:** 3
**Rating:** 4
**Confidence:** 4

**Summary:**

This paper introduces SAGP, a bio-inspired architecture that generalizes the classic MLP into a graph-structured neural network with dynamic topology. Unlike traditional MLPs with fixed layers and widths, SAGP features neuron growth, synaptic pruning, and apoptosis, enabling it to self-organize from a minimal initial state.

**Questions:**

1. Can you provide more comprehensive or systematic analysis of training cost?
2. How does SAGP compared to spiking neural network (SNN) in interms of efficiency? SAGP must be trained from scratch, while SNN, another brain-inspired efficient model can be distilled or converted from pretrained ANNs, enabling much faster convergence and practical reuse.

**Ethical Concerns:**

["NO or VERY MINOR ethics concerns only"]

**Final Justification:**

I have raised my confidence as I gave positive score in the review.

**Limitations:**

I would suggest to conduct more applicable experimental settings on larger datasets with training cost comparison.

**Quality:**

3

**Strengths And Weaknesses:**

Pros:
1. The idea of reformulation of MLPs as graph perceptrons is intersting.
2. The authors derive a clean algebraic formulation for MLPs using message passing on adjacency matrices. This lays a principled foundation for extending the MLP to non-layered, dynamic topologies.
3. The paper provides quantitative evidence of efficiency during inference. For example, on the FSDD dataset, SAGP achieves >99% of MLP performance using only ~0.7% of the synapses.

Cons:
1. All benchmarks are limited to small or medium datasets. The model is not tested on large-scale tasks (e.g., ImageNet), so scalability and training stability in such settings remain unverified.
2. SAGP lacks mechanisms to initialize or transfer topology across tasks (for the potential of building foundation model). Each run builds a new network from a minimal seed, limiting its applicability in real-world multitask or continual learning scenarios, and making it vulnerable to instability or variance across runs.

---

> ### Author Rebuttal · Authors · 2025-07-31
>
> We appreciate the reviewers' diligent efforts in providing thoughtful comments. Below is our response.
>
> # Cons 1 & Cons 2
> We would like to answer both cons simultaneously. First, we apologize that, given our limited time and resources, it is difficult for us to train a SAGP (or any other model containing SAGP) from scratch on the ImageNet dataset of 10 million images. However, we would like to emphasize that **SAGP can actually be initialized in non-minimal seed topologies.**  Note that the growth/death mechanism of neurons and synapses described in the paper is actually completely independent of the initialization. Therefore, we can initialize SAGP with any topology and weights, which is no difference from initializing with a minimum seed. Furthermore, we conducted the following experiments in the hope of addressing your concerns about both cons 1 and 2.
>
> First, we used the pre-trained weights of MobileNetV2 on ImageNet-1K, available in PyTorch. We then froze the network's backbone weights and used the weights of the final fully-connected layer as the initialization for the SAGP. (Specifically, the SAGP was initialized to a network with 1280 input neurons, 1000 output neurons, and 0 hidden neurons. The synaptic weights between each input-output neuron pair remained the same as those in the corresponding fully-connected layer of the original network.) We fixed the batch size to 128 and continued training for 100 full epochs (which required about one million backpropagations). The results are as follows:
>
> |                  | #Synapse | ACC   |
> | ---------------- | -------- | ----- |
> | MobileNetV2      | 1280000  | 71.8% |
> | MobileNetV2+SAGP | 435024   | 68.2% |
>
> Due to time constraints, we did not perform more epochs. However, according to our observations, performing more rounds would further reduce the number of synapse in SAGP. which is consistent with our conclusions in the original paper.
>
> # Question 1
> Of course, we now provide a theoretical and experimental analysis of the training cost of SAGP, and a detailed comparison with MLP.
>
> For simplicity, we ignore the activation function and bias in MLP/SAGP because they are not the main factors affecting time/memory cost. At this point, a two-layer MLP can be expressed as:
>
> $$
> \boldsymbol{z}=\boldsymbol{W}^T_2\boldsymbol{W}^T_1\boldsymbol{x}
> $$
>
> Where $\boldsymbol{x} \in \mathbb R^{|\mathcal I|}$,  $\boldsymbol{W}_1^T \in \mathbb R^{|\mathcal H| \times |\mathcal I|}$,$\boldsymbol{W}_2^T \in \mathbb R^{|\mathcal O| \times |\mathcal H|}$ and $\boldsymbol{z} \in \mathbb R^{|\mathcal O|}$。Recall that the time complexity of multiplying a $(n\times m)$ matrix by a $(m \times p)$matrix is $\Theta(nmp)$ and the space complexity is $\Theta(1)$. Now, we multiply $\boldsymbol{W}_1^T$ by $\boldsymbol{x}$, and then multiply the result by $\boldsymbol{W}_2^T$.
>
> After these two steps, the time complexity is $\Theta(|\mathcal H|\times|\mathcal O| + |\mathcal H|\times|\mathcal I|)$, and the space complexity is $\Theta(1)$ (We only consider the overhead beyond input and output). Note that the total number of synapses $S_{mlp}$ in this MLP is also $|\mathcal H|\times|\mathcal O| + |\mathcal H|\times|\mathcal I|$, so the time complexity can also be expressed as $\Theta(S_{mlp})$.
>
> Then we consider GP. For a GP that has undergone two message-passing, it can be simplified as:
>
> $$
> \boldsymbol{z} = \left((\boldsymbol{A}^T)^2[ \boldsymbol{x} \\ 0_{|\mathcal H| + |\mathcal O|}]^T  \right)_{[-|\mathcal O|:]}
> $$
>
>
>
> The definition of $\boldsymbol A$ is consistent with that of Formula 6 in the paper. GP use sparse matrix multiplication in its implementation, which in **once sparse matrix multiplication**, the time complexity and space complexity are both $\Theta(S)$, where $S$ is the number of non-zero elements in $\boldsymbol{A}$. Note that the number of non-zero elements of $\boldsymbol{A}$ is also equal to the number of synapses in the GP ($S_{gp}$), so for a GP with two message-passings, the time complexity and space complexity are $\Theta(2S_{gp})$ and $\Theta(S_{gp})$ respectively.
>
> The above conclusions can be extended to any layer of $L$-layer MLP or any $L$-times message-passing GP. Now, we Let the batch size is $B$, and the MLP and GP training epochs are $k_{mlp}$ and $k_{gp}$ respectively, then the total training complexity comparison is:
>
> |      | Time                        | Space               |
> | ---- | --------------------------- | ------------------- |
> | MLP  | $$\Theta(k_{mlp}BS_{mlp})$$ | $$\Theta(B)$$       |
> | GP   | $$\Theta(k_{gp}BLS_{gp})$$  | $$\Theta(BS_{gp})$$ |
>
> **Here, we would like to ask you to pay special attention to one detail**: in the table, it seems that the time complexity of GP is one factor $L$ higher than that of MLP, but in fact, when $\Theta(k_{mlp})=\Theta(k_{gp})$, the complexity of the two is equivalent. This is because $S_{mlp}$ increases linearly with $L$, but $S_{gp}$ is constant with $L$. In other words, $\Theta(S_{mlp})=\Theta(LS_{gp})$.
>
> For the complexity of SAGP, we only need to replace $S_{gp}$ in SP with $\max S_{sagp}$.
>
> We then give the following observation in our experiment:
>
> |                   | Photo  | Computers | ESC-50 | FSDD   | Fashion-MNIST | CIFAR-10 |
> | ----------------- | ------ | --------- | ------ | ------ | ------------- | -------- |
> | $S_{mlp}$         | 385536 | 659968    | 379904 | 883712 | 668672        | 920064   |
> | $$L\times\max S_{sagp}$$ | 30126  | 31257     | 66519  | 47565  | 184572        | 625923   |
>
> Based on the above analysis, we believe that, **theoretically, the (average) single-epoch training time complexity of SAGP is consistent with that of MLP. However, in practice, SAGP's higher time cost stems from the use of more epochs, while its higher space cost stems from the use of sparse matrix multiplication, which increases linearly with $S_{gp}$.**
>
> Next, we report the experimental results
>
> |                             | Photo         | Computers     | ESC-50        | FSDD          | Fashion-MNIST(batch_size=128) | CIFAR-10 (batch_size=128) |
> | --------------------------- | ------------- | ------------- | ------------- | ------------- | ----------------------------- | ------------------------- |
> | Training epoch (MLP)        | 1000          | 1000          | 1000          | 1000          | 1000                          | 1000                      |
> | Training epoch (SAGP)       | 50000         | 50000         | 50000         | 50000         | 100000                        | 100000                    |
> | Ratio                       | $\times$50    | $\times$50    | $\times$50    | $\times$50    | $\times$100                   | $\times$100               |
> | Training Time (MLP)         | 11s           | 16s           | 12s           | 13s           | 39s                           | 43s                       |
> | Training Time (SAGP)        | 754s          | 748s          | 634s          | 674s          | 3705s                         | 4031s                     |
> | Ratio                       | $\times$68.54 | $\times$46.75 | $\times$52.87 | $\times$56.24 | $\times$95.00                 | $\times$93.74             |
> | Training Peak memory (MLP)  | 734 Mb        | 636 Mb        | 651 Mb        | 898 Mb        | 512 Mb                        | 510 Mb                    |
> | Training Peak memory (SAGP) | 7020 Mb       | 14610 Mb      | 4994Mb        | 7833Mb        | 14189 Mb                      | 15530 Mb                  |
> | Ratio                       | $\times$9.56  | $\times$22.97 | $\times$7.67  | $\times$8.27  | $\times$27.71                 | $\times$30.45             |
>
> Please note that the number of SAGP training epochs shown in this table is not strictly necessary. In fact, reducing this number of epochs to 20% does not significantly reduce the model accuracy. We set this large number of epochs to explore the limits of synapse compression.
>
> We also acknowledge in the limitation section in the appendix that improving training efficiency is one of the core directions for optimizing SAGP. In addition, we discussed some methods for reducing the training complexity of SAGP in **sections 1.1 to 1.3 of our response to reviewer 4y96**. If you are interested, you are welcome to read them further.
>
> # Question 2
>
> SNN and self-assembling neural networks are both biologically inspired architectures, but their efficiency advantages are actually reflected in different aspects. SNN focuses on reducing energy consumption. Its core idea is to reduce unnecessary neuronal activities by realizing asynchronous triggering of neurons based on events, thereby reducing power consumption. The core of self-assembling networks lies in optimizing the connections between neurons to achieve a sparser topology. In other words, the efficiency of SAGP is reflected in the significant reduction in the number of parameters. Therefore, there is no strong comparison between the two.
>
> In addition, as we mentioned above, SAGP can actually use any MLP module in the trained model as its initialization and further train it. And this can indeed accelerate the convergence of the model. In particular, SAGP may be more suitable for continuous learning than MLP or SNN, because SAGP can learn new content by growing new neurons, thereby reducing the modification of the original weights.
>
> To verify this, we also added the following experiments: First, we split the 10 classes in the Fashion-Mnist dataset into two parts. We initially trained SAGP on data from only the first five classes. After the topology stabilized, we restarted the self-assembly process on data from the last five classes. At this point, we have the following results:
>
> |                           | MLP            | SAGP            |
> | ------------------------- | -------------- | --------------- |
> | ACC on first five classes | 90.7% -> 43.6% | 89.2% -> %61.7% |

---

> > ### Comment · Reviewer_v1ys · 2025-08-06
> >
> > Thanks for the response, I will maintain my positive score.

---

### Comment · Area_Chair_m4xC · 2025-08-04

Dear Reviewers,

The authors have submitted their responses to your reviews. At this time, I have not seen any replies in the discussion thread. Please take a moment to read the author responses and participate in the discussion during this period.

Your engagement is important to ensure a fair and thorough review process. Early participation is appreciated and allows for a more meaningful exchange.

Thank you very much for your time and contributions.

Best regards,

AC

---

### Decision · Program_Chairs · 2025-09-17

**Decision:**

Accept (spotlight)

**Comment:**

This paper proposes Self-Assembling Graph Perceptrons (SAGP), a novel architecture that generalizes MLPs into graph-structured networks with neuron growth, synaptic pruning, and apoptosis. The work is both theoretically rigorous by showing that MLPs are a special case of SAGP and empirically supported, with experiments demonstrating that SAGP achieves comparable accuracy to MLPs using only a fraction of the synapses. The approach also highlights advantages for interpretability and continual learning, making it a timely and creative contribution to biologically inspired architectures.

The main concerns raised by reviewers centered on training efficiency and scalability. While per-epoch complexity is theoretically comparable to MLPs, SAGP requires many more epochs and significantly more memory during training, raising questions about its applicability to large-scale datasets. Reviewers also noted the limited evaluation scope (mostly small- and medium-scale classification benchmarks) and the absence of comparisons to state-of-the-art architectures such as CNNs. These limitations prevent the work from being viewed as fully established for large-scale use cases.

The authors provided a thorough and convincing rebuttal, including new results on ImageNet-1K with MobileNetV2 initialization, detailed training/inference cost tables, and strategies to reduce complexity (e.g., MLP initialization, limiting neuron in-degree). Reviewers found these clarifications strong, with two upgrading their recommendations to strong accept. While efficiency and evaluation scope remain open challenges, the paper introduces a principled and impactful idea that will spark further research. Therefore, I recommend acceptance.